# Buccal Resveratrol Delivery System as a Potential New Concept for the Periodontitis Treatment

**DOI:** 10.3390/pharmaceutics13030417

**Published:** 2021-03-20

**Authors:** Magdalena Paczkowska-Walendowska, Jakub Dvořák, Natalia Rosiak, Ewa Tykarska, Emilia Szymańska, Katarzyna Winnicka, Marek A. Ruchała, Judyta Cielecka-Piontek

**Affiliations:** 1Department of Pharmacognosy, Poznan University of Medical Sciences, Święcickiego 4, 60-781 Poznań, Poland; mpaczkowska@ump.edu.pl (M.P.-W.); nrosiak@ump.edu.pl (N.R.); 2Department of Chemical Engineering, University of Chemistry and Technology Prague, Technická 3, 166 28 Prague, Czech Republic; dvorak492@gmail.com; 3Zentiva k.s., U Kabelovny 130, 102 37 Praha, Czech Republic; 4Department of Chemical Technology of Drugs, Poznan University of Medical Sciences, Grunwaldzka 6, 60-780 Poznań, Poland; etykarska@ump.edu.pl; 5Department of Pharmaceutical Technology, Faculty of Pharmacy, Medical University of Białystok, Mickiewicza 2c, 15-222 Białystok, Poland; emilia.szymanska@umb.edu.pl (E.S.); kwin@umb.edu.pl (K.W.); 6Department of Conservative Dentistry and Endodontics, Poznan University of Medical Sciences, Bukowska 70, 60-812 Poznań, Poland; maruchala@ump.edu.pl

**Keywords:** resveratrol, cyclodextrins, buccal tablets, mucoadhesion

## Abstract

The health benefits of resveratrol have been proven to inhibit the development of numerous diseases. A frequent limitation in its use is a low bioavailability stemming from a poor solubility and fast enterohepatic metabolism. Thus, the aim of the research was to investigate the possibility to formulate mucoadhesive cyclodextrin- and xanthan gum-based buccal tablets in order to increase the solubility of resveratrol and to eliminate bypass enterohepatic metabolism. Systems of resveratrol with α-cyclodextrin (α-CD), β-cyclodextrin (β-CD), γ-cyclodextrin (γ-CD) and 2-hydroxypropyl-β-cyclodextrin (HP-β-CD) prepared by the dry mixing method (ratio 1:1) were selected for the of tablets where xanthan gum was used as a mucoadhesive agent. They were identified on the basis of PXRD, FT-IR analysis. Tablets F1 (with α-CD), F2 (with β-CD) and F3 (with γ-CD) were characterized by the highest compactibility as well as by favorable mucoadhesive properties. Resveratrol release from these tablets was delayed and controlled by diffusion. The tablets prepared in the course of this study appear to constitute promising resveratrol delivery systems and are recommended to increase the effectiveness of the treatment in many diseases, particularly periodontitis.

## 1. Introduction

The periodontal diseases are highly prevalent and can affect up to 90% of the worldwide population [1]. Gingivitis, the mildest form of periodontal disease, is caused by the bacterial biofilm (dental plaque), e.g., *Porphyromonas gingivalis, Streptococcus mutans* as well as *Candida albicans*, that accumulates on teeth adjacent to the gingiva [2]. In addition to mentioned pathogenic microorganisms in the biofilm, genetic and environmental factors, especially antioxidant stress and chronic inflammation, are the cause of these diseases. Among the solutions available on the pharmaceutical market, both a few substances of natural origin and synthetic compounds can be used. An extremely interesting idea is the use of herbal substances because they can combine antioxidant, anti-inflammatory and antibacterial properties [3]. One of these substances is resveratrol.

Resveratrol (3,5,4′-trihydroxytrans-stilbene, RSV) is a compound belonging to the stilbene group. In plants, it plays the role of phytoalexin, which is produced in response to the infection caused by the *Botyrytis cinera* pathogen, and as a result of adverse environmental conditions, such as oxidative stress, exposure to sunlight, or heavy metals [4]. Modern research papers have shown that the biological and pharmacological activities of resveratrol are related to its potent antioxidant properties, as well as anti-inflammatory activity and effects on lipid metabolism. In fact, the antioxidant and anti-inflammatory activity (inhibition of NF-kappa B in LPS, TNF-α), associated with antimicrobial activity (against, e.g., *Streptococcus mutans*, known as cariogenic oral bacteria), can play a crucial role in the treatment of periodontitis [5,6,7]. Moreover, resveratrol has also been demonstrated to inhibit platelet aggregation [8] and to have an antiestrogenic activity [9,10]. Numerous in vivo studies have indicated its high anti-cancer, chemopreventive, and chemotherapeutic potential. In addition, the anti-diabetic and neuroprotective activity of stilbene have also been established [11,12]. It was demonstrated that antineoplastic properties are related to the structure of resveratrol, whereas the 4-hydroxyl group in trans-conformation (hydroxyethyl moiety) is responsible for its antiproliferative activity [13].

According to the Biopharmaceutical Classification System (BCS), resveratrol is classified as a second-class substance which means it is characterized by low solubility in water, although it possesses a high membrane permeability [14]. Low solubility is the first factor resulting in the low resveratrol bioavailability. Orally administered resveratrol is easily absorbed from the small intestine due to its non-polar nature and small molecular structure [15]. Despite rapid systemic absorption, resveratrol achieves low bioavailability below 1% due to high-throughput metabolism, mainly in enterocytes and the liver. [16,17,18]. Conjugations of glucuronide and sulphate are the major metabolic pathways identified in humans [18,19]. As a consequence of rapid enterohepatic metabolism, the half-life of trans-resveratrol was estimated at 1–3 h following single-doses and 2–5 h following repeated doses [20].

Owing to a low solubility, it is necessary to search for release modifiers, as well as alternative routes of administration bypassing the enterohepatic metabolism. One of the developed methods to increase resveratrol bioavailability is cyclodextrins complexation. Cyclodextrins (CDs) are oligosaccharides that, due to their unique nature, can form inclusion complexes with different molecules [21]. Therefore, the key function of CDs is to increase the dissolution and bioavailability of poorly soluble drugs belonging essentially to the second class of BCS [22]. Hence, resveratrol seems to be the perfect model drug for such a purpose [23]. Moreover, CDs can be used to mask the bitter taste of drugs, which constitutes an additional valuable feature in terms of oral drug administration [24]. In fact, CDs can be used as non-complexing excipients of tablets, such as fillers, disintegrants, binders, and multifunctional direct compression excipients [22].

From the technological point of view, a route of administration that avoids the first-pass metabolism, whereby the drug would bypass the gastrointestinal tract, thus increasing the bioavailability of resveratrol and consequently allowing higher concentrations at active sites, has been thoroughly researched [25]. One of such ways is oral transmucosal drug delivery. The oral mucosa is highly vascularized; therefore, drugs absorbed through the oral mucosa pass directly to the systemic circulation, bypassing the gastrointestinal tract and the first-pass hepatic metabolism [26]. Such alternative rough of administration can increase resveratrol absorption and greatly reduce the inter-human variability of the peak plasma concentration and metabolite profile to nearly low in order to improve clinical utility [27]. In addition, oral transmucosal formulation can circumvent the physiological limitations of resveratrol absorption and bioavailability that often occur after tablet, food, or fluid administration [28]. Actually, resveratrol has high potential as a candidate for oral mucosal absorption, because of its beneficial properties such as relatively low molecular weight that remains uncharged at physiological pH, which allows passive diffusion through the buccal mucosa [26,29]. The preparation of a lozenge containing resveratrol was the goal of a study designed by Blanchard et al., which resulted in surprisingly very high peak plasma levels compared to that obtained for similar doses of resveratrol administered in a traditional way using the oral formulation [27]. Moreover, the mucoadhesive film with resveratrol, designed by Ansari et al., was considered for buccal application as well and it was found that the film did not induce any histopathological abnormalities in the goat buccal mucosa [30]. Another pharmaceutical dosage form, such as mucoadhesive tablets, was investigated as a potential treatment strategy for oral inflammatory lesions [31]. The bioavailability of resveratrol delivered through the oral mucosa may be over one log higher than by swallowing it, as determined by the fraction of the initial resveratrol intake in the blood and, in the metabolized form, in urine [32].

Given the possibility of increasing the bioavailability of resveratrol by mucoadhesive in the buccal delivery system, it is worth emphasizing the possibility of treating periodontitis by means of such delivery systems. Chronic periodontitis is an inflammatory disease that diminishes tooth-supporting structures and is dependent on neutrophil recruitment as well as oxidative stress. Since resveratrol blocks neutrophil recruitment and oxidative bursts, it can help effectively reduce periodontitis. Obtaining a mucoadhesive delivery system will extend the contact time of resveratrol with the with a diseased place within the oral cavity, and will result in a stronger clinical effect. Therefore, the aim of the studies was to investigate the possibility of formulating mucoadhesive cyclodextrin- and xanthan gum-based buccal tablets to increase resveratrol solubility and eliminate bypass the enterohepatic metabolism. Hence, it can be assumed that resveratrol administered in this form will achieve greater bioavailability than when administered in the form of conventional oral tablets, and it will show both a stronger and a more prolonged local effect.

## 2. Materials and Methods

### 2.1. Chemicals and Reagents

Resveratrol (98%) (RSV) isolated from the giant knotweed powder extract was supplied by PK Components (Warsaw, Poland). Trans-resveratrol (≥95.0%, reference substance), α-cyclodextrin (α-CD), β-cyclodextrin (β-CD), γ-cyclodextrin (γ-CD) and 2-hydroxypropyl-β-cyclodextrin (HP-β-CD), xanthan gum, as well as magnesium stearate were obtained from Sigma-Aldrich (Poznan, Poland). Potassium dihydrogen phosphate was supplied by Avantor Performance Materials Poland S.A. (Gliwice, Poland), Prisma™ HT buffer, Acceptor Sink Buffer, and GIT lipid solution were obtained from Pion Inc. (Billerica, MA, USA), whereas HPLC grade acetonitrile was obtained from Merck (Warsaw, Poland). High-quality pure water and ultra-high-quality pure water were prepared using an Direct-Q 3 UV Merck Millipore purification system.

### 2.2. Preformulation Studies

Preformulation steps comprised preparation of cyclodextrin systems and their spectroscopic (PXRD and FT-IR) characterization, in vitro dissolution behavior of the prepared systems and the permeation of RSV through the gastrointestinal tract parallel artificial membrane permeability assay (GIT PAMPA). The abovementioned analyses constitute a prerequisite for proceeding to the formulation by means of a quality assessment of the behaviour of the RSV/CD powders and a reduction in the number of samples.

#### 2.2.1. Preparation of the Cyclodextrin Systems

Solid inclusion complexes of RSV with α-CD, β-CD, γ- CD and HP-β-CD were prepared by three different methods [33]:–Method 1 (dry mixing, DM)—RSV and CD starting material powders in a molar ratio of 1:1 and 1:2 were added to an agate mortar and pestle. The materials were subjected to a dry mechanochemical activation for 60 min.–Method 2 (kneading with an ethanol/water mixture, Kn)—RSV and CD in a molar ratio of 1:1 and 1:2 were added to an agate mortar and pestle and kneaded with an ethanol-water (1:3 *v*/*v*) mixture until the solvent evaporated.–Method 3 (solvent evaporation, Evap)—The aqueous solution of CD was added to an ethanol solution of RSV (in an RSV/CD molar ratio of 1:1 and 1:2). The mixture was evaporated using Rotavapor^®^ R-300 (Buchi) at 45 °C until dry.

#### 2.2.2. Identity Study of Solid Samples

##### Powder X-ray Diffraction Characterization (PXRD)

PXRD characterization was performed at room temperature by using a Rigaku Miniflex II, desktop X-ray diffractometer (Rigaku, Tokyo, Japan) equipped with a Cu Kα radiation X-ray source and a Haskris cooler (Haskris, Elmhurst, IL, USA). The samples were scanned over a range of 5–40° 2*θ* with a step width of 0.05° 2*θ* and signal collection time of 1 s per step.

##### Fourier Transform Infrared Spectroscopy (FT-IR)

The formation of the RSV/CD systems was confirmed using the FT-IR method. FT-IR measurements were conducted at room temperature using a Fourier transform infrared (FT-IR) spectrometer, Bruker Equinox 55 spectrometer (Bruker Optics, Ettlingen, Germany). All systems (RSV/α-CD, RSV/β-CD, RSV/HP-β-CD, RSV/γ-CD) were prepared in a KBr matrix in a ratio of 1 mg sample per 200 mg KBr. All measurements were performed at room temperature with the following parameters: resolution—4 cm^−1^, number of scans—400, wavenumber range—400–4000 cm^−1^, blank—pure KBr pellet.

##### High-Performance Liquid Chromatography with Diode-Array Detection (HPLC-DAD) Method Development and Validation

The RSV concentrations were determined with the HPLC-Diode-Array Detection method. The separation of trans-RSV in the presence of its impurities, i.e., cis-RSV, was possible using the Shimadzu Prominent-i LC-2030C (Shimadzu, Kioto, Japan). Kinetex-C18 column (100 × 2.1 mm, 5.0 μm) was used as a stationary phase (Phenomenex, Warsaw, Poland), while a mobile phase was composed of 0.5% acetic acid (pH 2.93) and acetonitrile (80:20 *v*/*v*) with flow rate of 1.0 mL min^−1^. A diode array detector was set at a wavelength maxima (λ_max_) of 306 and 285 nm. The column was set at 40 °C.

The HPLC-DAD method was validated according to the International Conference on Harmonization Guideline Q2. It comprised specificity and selectivity, linearity, range of linearity, intra- and inter-day precision, limits of detection (LOD) and quantitation (LOQ), and robustness.

#### 2.2.3. Evaluation of Pharmaceutical Properties of Solid Samples

##### Dissolution Studies

Dissolution studies were performed with an Agilent 708-DS dissolution apparatus (Agilent, Santa Clara, CA, USA). A standard paddle method was used at 37 ± 0.5 °C with a stirring speed of 50 rpm. RSV and its systems with CDs were weighed into gelatine capsules and placed in a sinker to avoid capsule flotation on the liquid surface. The systems were placed in 900 mL of the phosphate buffer at pH 6.8. The liquid samples were collected at the specified time intervals, an equal volume of temperature-equilibrated media was replaced, and obtained samples were filtered through a 0.45 μm nylon membrane filter. The concentrations of RSV in the filtered solutions were determined by the HPLC method as described above.

The dissolution rate profiles were compared by using two-factor values (*f*_1_ and *f*_2_) model proposed by Moore and Flanner [34], using following equations:(1)$$f_{1}=\frac{\sum_{j=1}^{n}\left|R_{j}-T_{j}\right|}{\sum_{j=1}^{n}R_{j}}\times 100$$
(2)$$f_{2}=50\times \log\left({\left(1+\left(\frac{1}{n}\right)\overset{n}{\underset{j=1}{\sum }}{\left|R_{j}-T_{j}\right|}^{2}\right)}^{-\frac{1}{2}}\times 100\right)$$
where *n* is the number of samples, *R*_j_ and *T*_j_ are the percentages dissolved of the reference substance (RSV) and the tested system (RSV/CD systems) at each time point *j*. Dissolution profiles are considered similar when the *f*_1_ value is nearly close to 0 and *f*_2_ to 100.

##### Permeability Studies

The permeability of RSV and RSV-CD systems was investigated using a parallel artificial membrane permeability assay simulating the gastrointestinal tract environment (PAMPA GIT), according to the methodology described by Paczkowska et al. [33]. The samples were dissolved in donor solution at pH 6.8. The plates were incubated at 37 °C for 3 h in a humidity-saturated atmosphere. The RSV concentrations changes in both the donor and acceptor compartments were measured by HPLC-DAD method, described in Section “High-Performance Liquid Chromatography with Diode-Array Detection (HPLC-DAD) Method Development and Validation”.

The apparent permeability coefficient (*P_app_*) was calculated from the following equation:(3)$$P_{app}=\frac{-ln\left(1-\frac{C_{A}}{C_{equilibrium}}\right)}{S\times \left(\frac{1}{V_{D}}+\frac{1}{V_{A}}\right)\times t}$$
where *V_D_*-donor volume, *V_A_*-acceptor volume, *C_equilibrium_*-equilibrium concentration ($C_{equilibrium}=\frac{C_{D}\times V_{D}+C_{A}\times V_{A}}{V_{D}+V_{A}}$), *C_D_*-donor concentration, *C_A_*-acceptor concentration, *S*-membrane area, *t*-incubation time (in seconds) [35].

To verify whether *P_app_* determined for permeability was statistically different, an ANOVA test using Statistica 12.0 software was employed. On the basis of the test, it was established that compounds with *P_app_* < 1 × 10^−6^ cm s^−1^ are classified as low-permeable, whereas those with *P_app_* > 1 × 10^−6^ cm s^−1^ are categorized as high-permeable compounds [36].

##### Antioxidant Activity—DPPH and CUPRAC Assays

The DPPH assay was performed as follows: to 25 μL of RSV (concentration range 25–400 μg mL^−1^), a 175 μL DPPH (2,2-diphenyl-1-picryl-hydrazyl-hydrate) solution (0.078 mg mL^−1^ in methanol) was added, obtaining the final assay concentrations in range 3.1–50.0 μg mL^−1^. The plate with reaction mixture was shaken for 5 min at 600 rpm and incubated for next 30 min at room temperature in dark conditions. The absorbance at λ = 517 nm was examined against the blank (25 μL DMSO with 175 μL MeOH). Additionally, 25 μL DMSO with with 175 μL DPPH solution was obtained as control sample. The DPPH scavenging activity was calculated using the following equation:(4)$$DPPH\;scavenging\;activity\;\left(\%\right)=\frac{A_{0}-A_{1}}{A_{0}}*100\%$$
where *A*_0_ is the control sample absorbance, *A*_1_ is the RSV sample absorbance [37].

All analyses were repeated six times. The results were expressed as the *IC*_50_ value, corresponding to the RSV or RSV/CD system concentration required to inhibit DPPH radical formation by 50% and was determined by using the quadratic equation.

The cupric reducing antioxidant capacity (CUPRAC) assay was conducted according to the guidelines of Apak et al. with certain modifications [38]. The solutions of the CUPRAC reagent included equal parts of 7.5 mM neocuproine solution in 96% ethanol, acetate buffer (pH = 7.0), and 10 mM CuCl_2_·H_2_O solution. Subsequently, to 50 μL of RSV (concentration range 25–400 μg mL^−1^), 150 μL of CUPRAC solution was added to obtained the final assay concentrations at range 6.2–100 μg mL^−1^, and then mixed and incubated for 30 min at room temperature in the dark condition. Then the absorbance was read at λ = 450 nm. The analysis was performed in six replicates. The results were expressed as the *IC*_0.5,_ which corresponds to the extract concentration required to obtain the absorbance value of 0.5.

To verify whether *IC*_50_ and *IC*_0.5_ determined in antioxidant activity assays were statistically different, an ANOVA test using Statistica 12.0 software was applied.

### 2.3. Formulation Studies

Binary systems of RSV/CD systems and the following excipients: xanthan gum and magnesium stearate in weight ratio 1:1 (*w*/*w*) were obtained. Extra binary systems containing RSV/CD systems and excipients in a weight ratio from the designed formulation were also prepared (Table 1). Binary systems passed the stability tests at room temperature under controlled air humidity at RH = 50%. At defined time points (3, 6 and 12 months), the RSV concentrations changes were determined by the HPLC-DAD method.

Moreover, the formulation composition in powder systems (Table 1) was tested to confirm that the antioxidant properties were maintained (according to the methodology in Section “Antioxidant Activity—DPPH and CUPRAC Assays”).

#### 2.3.1. Tableting Process

Flat-faced, 8 mm in diameter, tablets were compressed using a laboratory scale, single punch tableting machine, NP-RD10A Tablet Press (Natoli, Saint Charles, MO, USA). Compaction properties of tablets were assessed using a number of various compaction forces in the range of 2.5 to 10 kN. The pressure was released when the desired compaction pressure was achieved. The composition of the tablets is presented in Table 1.

#### 2.3.2. Tablet Characterization

The freshly produced tablets was weighted immediately after their compaction. The tablet mass uniformity was controlled on the basis of the method described in Ph.Eur. 9th [39]. Additionally, the thickness and the diameter of 20 randomly selected tablets were measured by using a manual vernier caliper. After all measurements, mean values and standard deviations were calculated (SD).

The tablet hardness was established according to the methods described in Ph.Eur. 9th, and was assessed using the PTB-M manual tablet hardness testing instrument (Natoli, Saint Charles, MO, USA). Each hardness value is an average of six measurements and is expressed as a mean value with a SD.

Tensile strength (*σ*) values were calculated on the basis of the breaking force (*F*) values (*N*), where d is the diameter of the tablet (mm) and h is the thickness of the tablets (mm) [40].
(5)$$\sigma =\frac{2F}{\pi dh}$$

Solid fraction (*SF*) was calculated by the equation, where *W_t_* is the weight of tablet (mg), *v* is the tablet volume, *ρ_true_* is the powder true density (g/cm^3^).
(6)$$SF=\frac{W_{t}}{\rho_{true}v}$$

The tablet porosity (*ε*) was calculated from *SF* using the following equation:(7)ε=1−SF

#### 2.3.3. In Vitro Release Studies

The study was performed according to the method described in Section “Dissolution Studies” with some modifications. Dissolution studies were performed with an Agilent 708-DS dissolution apparatus (Agilent, Santa Clara, CA, USA). A standard paddle method was used at 37 ± 0.5 °C with a stirring speed of 50 rpm. Tablets were placed in 50 mL of the phosphate buffer at pH 6.8. The liquid samples were collected at the specified time intervals, an equal volume of temperature-equilibrated media was replaced, and the samples were filtered through a 0.45 μm nylon membrane filter. The concentrations of RSV in the filtered solutions were determined by the HPLC method as described in Section “High-Performance Liquid Chromatography with Diode-Array Detection (HPLC-DAD) Method Development and Validation”.

The release profiles were compared by means of the model proposed by Moore and Flanner, which is based on two-factor values, *f*_1_ and *f*_2_ [34].

In order to investigate the release kinetics, the obtained active compounds release profiles were fitted to the following mathematical models [41]: zero-order equation: F=k×t, first-order equation: lnF=k×t, Higuchi equation: $F=kt^{1/2}$, Korsmeyer-Peppas equation: $F=kt^{n}$, where *F*-the fraction of release drug, *k*-the constant associated with the release, and *t*-the time.

#### 2.3.4. Mucoadhesive Properties

##### In Vitro Assessment of Mucin-Biopolymer Bioadhesive Bond Strength

A viscometric method was used to quantify mucin-chitosan bioadhesive bond strength. The assessment was performed according to the method described by Hassan and Gallo [42].

The viscosity coefficient of a hydrophilic dispersion containing mucin and a bioadhesive polymer was calculated from the equation provided below:
*η_t_* = *η_m_* + *η_p_* + *η_b_*(8)
where *η_t_* is the system viscosity coefficient, *η_m_* is mucin viscosity coefficients, *η_p_* is polymer viscosity coefficients, *η_b_* is the component viscosity due to bioadhesion, which is obtained by transforming the following equation:
*η_b_* = *η_t_* − *η_m_* − *η_p_*(9)

The bioadhesion force *F*, which represents the intermolecular frictional force per unit area, is calculated by following equation:*F* = *η_b_σ*(10)
where *σ* is the shear rate per second.

##### Determination of the Ex Vivo Mucoadhesive Properties (Maximum Detachment Force and Work of Mucoadhesion)

The mucoadhesive behavior of tablets in contact with the buccal mucosa excised from the porcine cheek was investigated using the tensile experiments on a TA-XT Plus texture analyzer (Stable Microsystems, Godalming, UK) equipped with the measuring system G/muc. The fresh porcine buccal mucosa was kindly provided from local slaughterhouse Bost (Turosn Koscielna, Poland). The tissue was a slaughterhouse waste prepared by a veterinarian. The experiment did not require the approval of bioethical or ethical committee. The fresh porcine buccal mucosa was used as the mucoadhesive membrane, while the simulated saliva fluid (SSF, pH 6.8) [43] served as the moistening medium. The tissue was attached with cyanoacrylate glue to the thermostated stainless steel plate and conditioned at 37.0 ± 2 °C for 5 min. Each tablet was then adhered with glue to the upper probe and wetted with 50 μL of SSF. The operating parameters selected for the experiments were as follows: a pre-test and a post-test speed 2 mm/s, contact time 60 s and a contact force 0.5 N. The maximum detachment force (mN) necessary to separate the tablet from the porcine buccal tissue was recorded directly from Texture Exponent 32 software, while the work of the mucoadhesion (expressed in J) was calculated from the area under the force curve as a function as distance. Cellulose paper constituted the negative control. The studies were performed at least in triplicate.

Data were expressed as mean ± standard deviation (SD) by MS Excel software. Measurements were considered significant at *p* < 0.05. Results from mucoadhesive studies were evaluated statistically by non-parametric Kruskal–Wallis test followed with Dunn–Bonferroni post hoc method with using Statistica 12.0 software. Shapiro–Wilk test was implemented to check data distribution normality.

##### Determination of the Residence Time

The residence time was determined using an adjusted apparatus for the disintegration time test, according to Nakamura et al. [44]. The medium was 500 mL of SSF pH 6.8 maintained at 37 ± 2°. The segments of porcine buccal mucosa, each 2–3 cm length, were adhered to the mucosal surface face upwards to the inner surface of a glass beaker. Each tablet was brought into contact with the tissue by applying a finger force for 10 s. Subsequently, a plexiglass cylinder (of 6 cm in diameter, weighing 280 g) was placed in the apparatus, and it moved up and down continuously as soon as the test commenced. During the measurements, each attached tablet was completely immersed in the medium at the highest point of the cylinder location and was out at the highest point of the cylinder location. The test lasted no longer than 3 h and the time necessary to detach formulation from the mucosal surface was recorded (*n* = 3).

Quantitative variables were expressed as mean ± standard deviation (SD) with the significance level at *p* < 0.05. Results were evaluated statistically by non-parametric Kruskal-Wallis test followed with Dunn-Bonferroni *post* hoc method with using Statistica 12.0 software; while Shapiro–Wilk test checked the normality of data distribution.

## 3. Results and Discussion

### 3.1. Preformulation Studies

Solid inclusion complexes of RSV with α-CD, β-CD, γ-CD and HP-β-CD were prepared for increasing physicochemical properties of the initial resveratrol, dissolution rate in particular. Solid complexes were prepared using three different methods, including the formation in the solid-state (Method 1), the formation in the semisolid state (Method 2) and the formation in solution (Method 3). All samples were initially analyzed by PXRD, and FT-IR, and the most appropriate method of formation was selected on the basis of the results indicating the largest degree of the solid-state changes.

The solid systems were evaluated using PXRD. It was revealed that RSV was crystalline in nature, as evidenced by the position of diffraction peaks at 16.6°, 19.4°, 22.4°, 23.7°, 25.5°, 28.5° 2*θ* (Figure 1) [45]. The α-CD, β-CD and γ-CD starting material powders were also crystalline, in contrast to HP-β-CD, which was X-ray amorphous (Figure 1). Furthermore, the complexes revealed RSV diffraction peaks; however, they were of low intensity (Figure 1), suggesting an introduction of a disorder into the samples. The greatest disappearance of the crystalline bands was observed for the systems in the 1:1 molar ratio prepared by the dry mixing method, which was selected for further investigation.

Against that, the RSV/HP-β-CD pattern had no diffraction peak corresponding to RSV, which suggests that the RSV no longer exists in a crystalline form when complexing with HP-β-CD within the evaporation process, although it exists in an amorphous state [46]. Additionally, since HP-β-CD is in higher energy amorphous state, the simple co-grinding mechanochemical activation process was enough to break the RSV crystal lattice, resulting in almost complete amorphization and complex formation. In fact, this is consistent with the solid-state inclusion complex formation mechanism, which required a particle size reduction and the development of crystal lattice defects, and then formation of complex on the reactants surface. To sum up, PXRD patterns confirmed the inclusion complex formation between CDs and RSV. Diffractograms for all obtained systems are presented in the Appendix A).

The powder RSV/CD systems were further characterized by FTIR. Due to the different number of glucose units in the cyclodextrin structures (α-CD, β-CD, HP-β-CD, γ-CD), slight differences were visible in the IR spectra of pure molecules (Figure 2). The changes are evident in such ranges as 500–900 cm^−1^, in which bending vibrations in the plane and outside of the O–H, C–H bonds as well as C–C and C–O stretching vibrations are observed. Moreover, shifts of the three most intense bands are visible for the characteristic bands related to the stretching vibrations of C–O and C–C bonds (1000–1200 cm^−1^). In the case of the HP-β-CD spectrum, special attention should be paid to the disappearance of the 998 cm^−1^ bands. This is due to the presence of the attached hydroxypropyl groups. The influence of these groups on the character of the spectrum is also visible around 3000 cm^−1^, where an additional component is registered at 2970 cm^−1^ at the band of 2928 cm^−1^ (vibration of the C–H bonds stretching).

Above 800 cm^−1^, the RSV/CD systems show bands derived from RSV (833 cm^−1^, C-H outside of the plane). In the range of 1200–1300 cm^−1^, two RSV bands are visible, i.e., 1247 cm^−1^ (CC stretching between hydroxyphenyl and ethenyl groups + CH rocking) and 1264 cm^−1^ (CO stretching in hydroxyphenyl group + CH rocking in hydroxyphenyl group). In terms of the pure substances in the systems, an additional band at 1513 cm^−1^ (C–O stretching + C–H rocking in hydroxyphenyl group) from RSV is visible. Additional components of 1588 cm^−1^ (C=C stretching + C–O–H bending in dihydroxyphenyl and hydroxyphenyl groups) and 1606 cm^−1^ (C=C stretching) are visible on the slope of the band around 1643 cm^−1^ and they are a combination of pure CD and RSV bands or may be a result of formation of new bond between RSV and CD. All FT-IR spectra are presented in Appendix A. Similar findings were observed by Kumpugdee-Vollrath et al., where new bands at 1595 and 1516 cm^−1^ were associated with new chemical bond formation between RSV and CD [47]. Also new band around 1592 cm^−1^ confirmed the formation of an authentic inclusion complex by Silva et al. [48]. The increased energy of stretching vibrations can be explained by the limitations of molecular motion related to the enclosure of the RSV into the CD cavity [48].

The identification and the quantification of trans-RSV contained in the raw material was performed by means of a high-performance liquid chromatography supported by a photodiode array detector (Figure 3). The retention time of the analyzed peak was compared with the retention time of the reference substance (trans-RSV) and its UV spectra.

The developed HPLC-DAD method was validated in accordance with the International Conference on Harmonization Guideline Q2. It comprised specificity and selectivity, linearity, range of linearity, intra- and inter-day precision, limits of detection (LOD) and quantitation (LOQ), and robustness. The validation parameters are shown in Appendix A.

The dissolution studies of the uncompressed powders were first performed to compare the changes in the dissolution rates of RSV from CD systems. RSV/CD showed more than two-fold (over 80%) higher drug release compared to RSV powder “as supplied” (38%) within 6 h (Figure 4). A complete dissolution of RSV occurred from all CD systems. Calculated *f*_1_ and *f*_2_ values confirmed that the dissolution profiles of RSV/CD systems are different from the pure RSV in the acceptor medium at pH 6.8.

The release profile of RSV/CD clearly indicated the high solubility of RSV which can be attributed to high solubilization and the subsequent amorphization following the encapsulation within CDs [49]. The best dissolution behaviour of RSV/HP-β-CD complex can be explained by the greatest water solubility, high amorphization, wetting, solubilizing and complexing power of this cyclodextrin [50]. Those findings are in line with previous RSV/CD solubility studies in which, first, a 1:1 ratio resulted in the complex formation, where inclusion ability of HP-β-CD was larger than that of β-CD. Secondly, HP-β-CD caused better solubilizing properties of RES [51], what was also visible in Figure 4, where the highest RSV release at 60 min was observed for the RSV/HP-β-CD system.

An in vitro permeability study through artificial membranes was performed with the use of the PAMPA GIT system. The apparent permeability (*P_app_*) value of RSV was (74.89 ± 17.28) × 10^−6^ cm s^−1^, according to the outlines of Yee, and thus was classified as a well-permeable compound [36]. Most cyclodextrin systems exhibited a lower permeability (Figure 5). In fact, a statistically higher permeability was achieved only for RSV/β-CD 1:1 Kn. The better permeability through artificial barriers of RSV/β-CD might stem from an increase in RSV solubility due to both the inclusion as well as non-inclusion phenomenon of the system, what was confirmed in Ansari et al. studies [52]. Permeability reduction in other cases confirmed that cyclodextrin systems’ nature is much more hydrophilic than pure RSV and, therefore, it is not so favoured by the passive diffusion through the gastrointestinal tract barrier [53].

The DPPH and CUPRAC test results confirmed a pronounced antioxidant activity of RSV (Appendix A; IC_50_ DPPH 22.1 μg mL^−1^; IC_0.5_ 33.2 μg mL^−1^). The research, aiming to compare the antioxidant activity, was subjected to the RSV system with various cyclodextrin types, prepared using several techniques. The system’s scavenging capability for the DPPH radicals, measured by IC_50_, ranged from 18.2 μg mL^−1^ (for RSV/γ-CD 1: 1 Evap) to 27.1 μg mL^−1^ (for RSV/β-CD 1: 1 Evap). Depending on the selected values, the RSV was similar. The RSV/HP-β-CD systems demonstrated the least activity, although it was still comparable to RSV.

In the case of the CUPRAC method, the IC_0.5_ antioxidant parameter determined for various RSV/CD systems ranged from 23.7 μg mL^−1^ to 34.6 μg mL^−1^. In contrast, RSV activity in the CUPRAC study was similar to the activity of the weakest systems (RSV/HP-β-CD and RSV/α-CD-*IC*_0.5_ > 30 µg mL^−1^). In the CUPRAC study, RSV was slightly less active than most tested samples. Significantly, the systems with RSV/γ-CD, regardless of the method following the RSV encapsulation, were characterized by a more potent activity than RSV itself.

Resveratrol activity was also tested in combination with all excipients within tablets. In this case, also no negative interaction with the excipient was demonstrated, and resveratrol antioxidant activity remained high (Appendix A).

Our results demonstrated that the tested RSV/CD samples did not differ significantly in the level of antioxidant activity when compared to the reference RSV. In general, the results indicated an increase in the activity of the complexes relative to the pure substance. However, more results suggesting an increase in the antioxidant activity concerned the CUPRAC method. Nevertheless, the activity tested can be considered high.

The previously performed studies displayed the differences in the antioxidant activity of the selected polyphenol/CD complexes, which are in agreement with the presented data. Compounds with the polyphenol structure: iridoids (oleuropein) [54], catechin derivatives (epigallocatechin gallate and gallocatechin gallate) [55], carotenoids (lycopene) [56] were also tested. The research most often concerned systems with β-CD less frequently than the other types, including HP-α-, dimethyl-b-CD, M-β-CD, HP-b-CD, SBE-β-CD, γ-CD, HP-γ-CD or α-CD. However, CDs were found to have poor antioxidant properties, which can increase the antioxidant potential of the active agents [57]. It also proves that the choice of cyclodextrin as a substance improving the physicochemical characteristics of poorly water-soluble resveratrol is correct since it improves the biological parameters of the compound, e.g., antioxidant properties as well. Lu et al. confirmed that RSV/HP-β-CD complex showed a higher antioxidant efficacy both in terms of capacity and rate of scavenging DPPH radical than RSV/β-CD complex [51]. The good safety profile is also demonstrated by the fact that cyclodextrins are used to prevent food browning and improve the antioxidant capacity of food [57].

### 3.2. Formulation Studies

Optimization of obtaining a mucoadhesive, buccal, pharmaceutical form containing the RSV/CDs systems was subsequently performed in order to obtain appropriate adhesion to the mucosa. Five systems (RSV/α-CD 1:1 DM, RSV/β-CD 1:1 DM, RSV/γ-CD 1:1 DM, RSV/HP-β-CD 1:1 DM and RSV/HP-β-CD 1:1 Evap) were selected since the production process in these systems was the most repetitive, as well as the most significant changes of RSV behavior were observed. Mucoadhesive formulations F1–F5 were prepared with a constant percentage of RSV (20.0 mg), CDs (1:1 molar ratio with RSV), xanthan gum (10% of RSV/CD system weight) which served as a mucoadhesive agent, whereas magnesium stearate (1% of final tablet weight) served as a lubricant (Table 1). Moreover, all excipients (CDs, xanthan gum and magnesium stearate) possess the GRAS status and can be used as food additives [58]. In fact, cyclodextrins in the tableting process, apart from the active compounds solubility modification function, can also be treated as fillers for the direct compression [22].

Tablet characterisation contained parameters such as tabletability, compressibility and compactibility (Figure 6). Tensile strength of the tablet, solid fraction and porosity at a range of compression pressure are the most essential parameters which describe the compaction properties of the investigated materials. Tabletability describes the capacity of the powder to be transferred into a tablet, and reflects the effect of increasing the compression force on the tablet’s tensile strength [59]. The tabletability of the tablets decreased in the following order: F3 > F1 > F2 > F4 > F5. The formulation F3 (RSV/γ-CD tablets) demonstrated the ability to produce the hardest tablets at low compaction pressures. Compering compressibility profiles, there no significant difference between F1–F5 formulations was observed. All prepared tablets can be characterized by a low porosity (Figure 6b). Whenever a pressure load is applied to powder sample, a decrease in its porosity or an increase in solid fraction value is observed; wherein both the porosity and solid fraction shape the compacted particles structure. The RSV/α-CD-based (formulation F1) and RSV/γ-CD-based (formulation F3) tablets showed higher compressibility, whereas a relatively high porosity was retained at higher compression pressure values. Powder compactibility is known as the capacity of a powder to form a coherent tablet in the course of the tableting process [59]. Additionally, it has been established that weaker tablets have greater porosity. This correlation between the aforementioned processes is due to a higher number of pores in the tablet resulting in a poor interparticle bonding and, therefore, the tablet requires a lower force in order to be broken down. The order of decreasing compactibility appears to be as follows: F3 > F1 > F2 > F4 > F5.

Based on the abovementioned parameters, RSV/α-CD-based (formulation F1) and RSV/γ-CD-based (formulation F3) tablets showed the best properties with a good tabletability, good compressibility, and also the highest compactibility, comparing to the other CDs systems. Tablets with RSV/HP-β-CD Evap systems did not meet the requirements for the compactibility.

The release kinetics of RSV from RSV/CD-based tablets were determined (Figure 7 and Appendix A). The differences in release kinetics, depending on the CD used, were observed. For all formulations F1–F5, the highest dissolution rate was observed for tablets with the highest compression pressure 150 and 200 MPa. It can be attributed to the fast disintegration of the tablets with the least compression pressure, the flotation of these particles, as well as to an insufficient moisture for further dissolution. All dissolution profiles were compared using *f_1_* and *f_2_* factor. Formulations F2, F4 and F5 were similar what confirmed similar structure and properties of β-CD and HP-β-CD used for tablets. To confirm that xanthan gum, beyond acting as mucoadhesive agent can prolong the RSV dissolution rate, tablets with xanthan gum and without cyclodextrin were prepared; the shapes of obtained release profiles resembled those presented in Figure 7, but RSV solubility was not greater than 60% after 360 min. On the basis of the dissolution profiles, it could be confirmed that all cyclodextrins, especially β-CD and HP-β-CD can act as a controlled-release excipient [60]. Moreover, HP-β-CD showed very similar properties as β-CD, predominantly at the level of the physics of compression and the drug release characteristics [24]. Therefore, the prolonged release of resveratrol from buccal tablets is the result of the use of cyclodextrin and xanthan gum in tablets composition.

To investigate the mechanism responsible for the resveratrol sustained release from tablets, the release data obtained for the formulation F1–F5 were fitted to the following dissolution models:zero-order, first-order equations, Higuchi model (used for the matrix systems), and the Korsmeyer-Peppas model (used for the swellable matrices) (Appendix A).

Two models of resveratrol which were the most probable, i.e., the release zero-order and the Higuchi model, were shown. They indicate that the release was not concentration-dependent, but it changed in time and occurred at an even rate. On the basis of the regression correlation coefficient, it could be established that RSV release from the tablets was controlled by diffusion in the Fickian diffusion mechanism, as this process was the most suitable for the Higuchi model. Therefore, diffusion seems to be one of the essential processes to release active compounds from xanthan gum-based buccal formulations [61].

Bioadhesive drug delivery systems have been designed to be located onto a biological surface. Cyclodextrin are also assigned mucoadhesive properties due to the hydrophilic outer part of the molecules, that are able to form hydrogen bonds with hydroxyl-groups on the sugars and other O- and N-containing groups on the protein backbone of the mucosa [62]. In formulations F1–F5, xanthan gum was added as a mucoadhesive polymer in the same ratio (10%), according to RSV/CD systems weight. As it is shown in Figure 8, there are huge differences in the bioadhesion components of each formulation, which can be caused by different cyclodextrin types. According to Gavini et al., HP-β-CD should possess a higher mucoadhesive capacity than β-CD due to its higher hydrophilicity [62]. The differences in our results could be caused by the addition of xanthan gum. Moreover, a combination of xanthan gum with β-CD improved the viscoelastic behavior of the biopolymer [63]. Mucoadhesive properties of the investigated formulation were arranged in the following order: F2 > F1 > F3 > F4 > F5.

Mucoadhesive behavior was also assessed by ex vivo measurements of the force required to separate each tablet from porcine buccal mucosa. Figure 9 presents the maximum detachment force and work of mucoadhesion of the tested formulations F1–F5. Regarding buccal delivery, the detachment force parameter might describe the rapid mechanical stress, e.g., resulting from the mouth movements disturbing the contact between formulation and mucosal tissue, whereas the work of mucoadhesion reflects the overall ability to retain in the application site.

Basically, all tablets were capable of adhering to the mucosal tissue which resulted from the presence of xanthan gum as a mucoadhesive agent. Certain alterations between the formulations were observed and tablets F1-F3 were found more adhesive as compared to F4-F5. This observation may suggest that the presence of HP-β-CD impairs the interaction between a mucoadhesive agent and the mucosal tissue. The formulation F1 (with α-CD) and F3 (with γ-CD) displayed the greatest values of mucoadhesiveness, most probably as a result of relatively high porosity (Figure 9) facilitating tablet wetting. This, in turn, favors the initial contact stage and further interpenetration of the mucoadhesive polymer and mucin chains. Interestingly, the method of cyclodextrin system preparation influenced the mucoadhesive capacity as the formulation F5 in which the solvent evaporation method was used exhibited higher values of the tested parameters in comparison to the formulation F4 prepared after dry mixing of the ingredients. The impact of compaction force on the mucoadhesive profile of tablets was also noticed. In general, formulations compressed with a lower compression pressure 150 MPa displayed a greater work of mucoadhesion, although slightly lower values of detachment force which might indicate their susceptibility to sharp stress e.g., from tongue movements.

Tablets were additionally subjected to the residence time test in order to investigate their mucoadhesive characteristic after subjecting to strains simulating mouth movements upon continuous contact with the saliva fluid simulant (Table 2). All investigated formulations adhered immediately to the porcine buccal mucosa. Upon gradual swelling, formulations F1, F2 and F5 remained in contact with the tissue over the period of 3 h. In contrast, the residence time of the formulation F4 was limited as a consequence of its relatively fast disintegration in the simulated saliva fluid regardless of the applied compaction force (Table 2). Due to the substantial weight increase associated with a great water uptake, the core of tablets F3 was dissected into two fragments within 120 min which was recognized as the endpoint of the test.

Based on all the obtained results, F2 containing RSV with β-CD turned out to be the most valuable. This formulation exhibited good tabletability, compressibility, as well as good compactibility, not the highest values. Despite the fact that F1 and F3 showed the best properties, too high tablet hardness turned out to be a limiting factor for the release of the active ingredient. In addition, F2 showed good mucoadhesive properties, allowing the tablet to remain at the site of administration for 3 h. This time, in turn, ensures that the RSV is released from the tablet in 70%. For formulations 1 and 3, RSV release at 180 min was 40%.

## 4. Conclusions

The buccal cyclodextrin/xanthan-based tablets containing trans-resveratrol with a rare application algorithm are a convenient alternative to the traditionally orally administered resveratrol products. The combination of the excipients ensures appropriate antioxidant mechanism of action. Based on all the obtained results, formulation F2 containing resveratrol with β-cyclodextrin turned out to be the most valuable, with good mechanical and dissolution properties as well as appropriate mucoadhesive properties that ensured prolonged operation at the application site. To conclude, the developed buccal formulations constitute good candidates for effective treatment of periodontitis due to the unique anti-inflammatory and antioxidant properties of resveratrol.

## Figures and Tables

**Figure 1 pharmaceutics-13-00417-f001:**
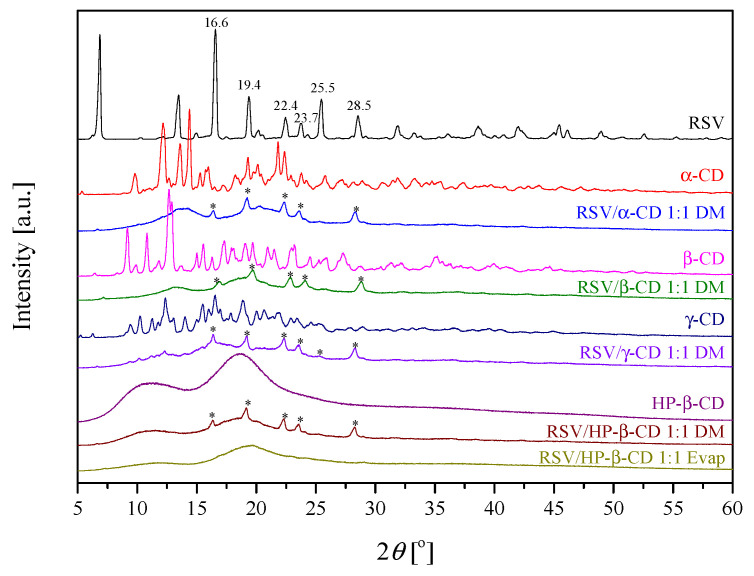
X-ray diffractograms of the solid samples. 2*θ* positions of the principal diffraction peaks are shown for RSV, whereas the traces of crystalline RSV peaks in the systems are indicated by “*”.

**Figure 2 pharmaceutics-13-00417-f002:**
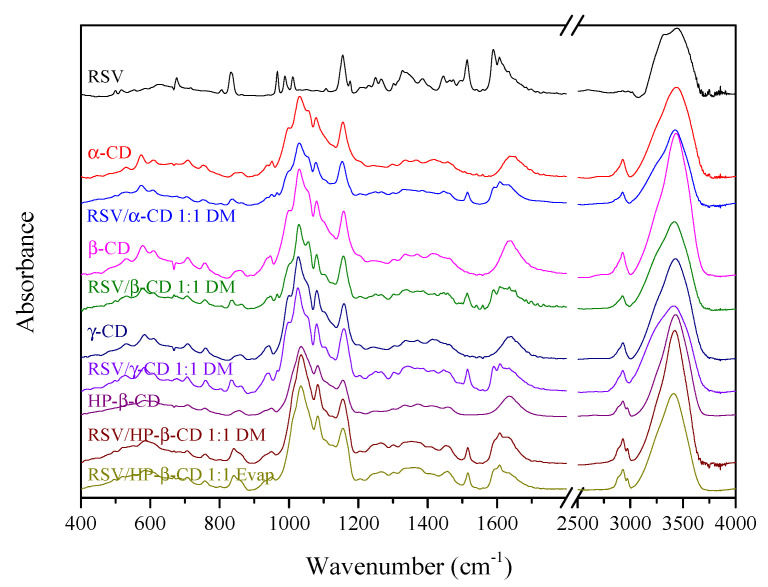
FT-IR spectra of selected powder resveratrol (RSV)/cyclodextrin (CD) systems.

**Figure 3 pharmaceutics-13-00417-f003:**
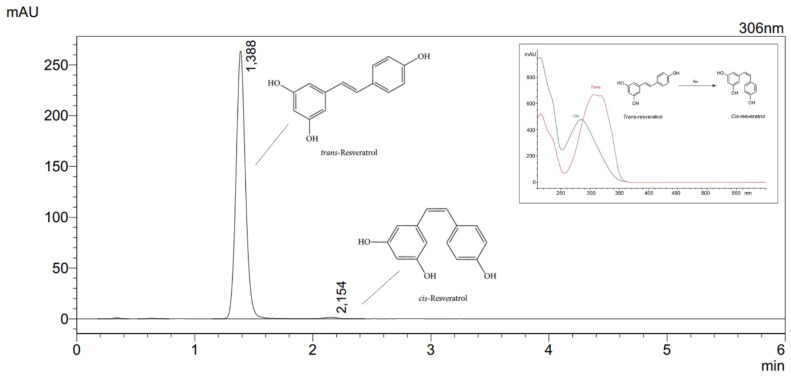
Chromatogram of RSV (c = 40 µg mL^−1^).

**Figure 4 pharmaceutics-13-00417-f004:**
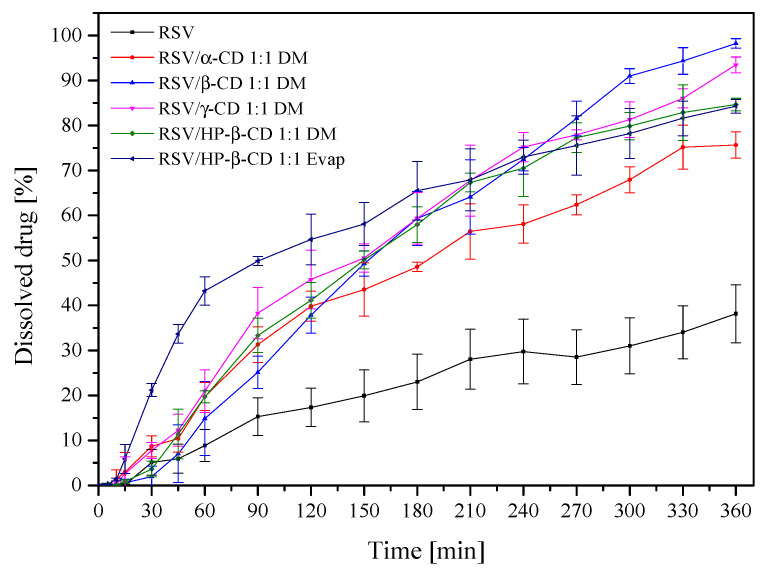
Powder dissolution of RSV from the powder CD systems at pH 6.8.

**Figure 5 pharmaceutics-13-00417-f005:**
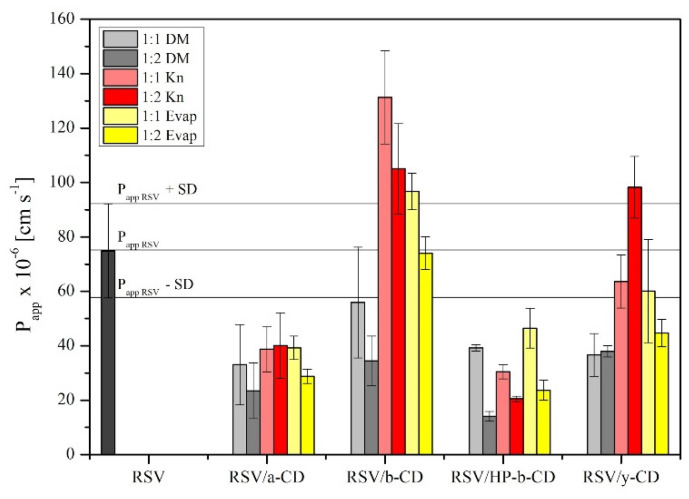
The permeability values of pure RSV and its CDs systems.

**Figure 6 pharmaceutics-13-00417-f006:**
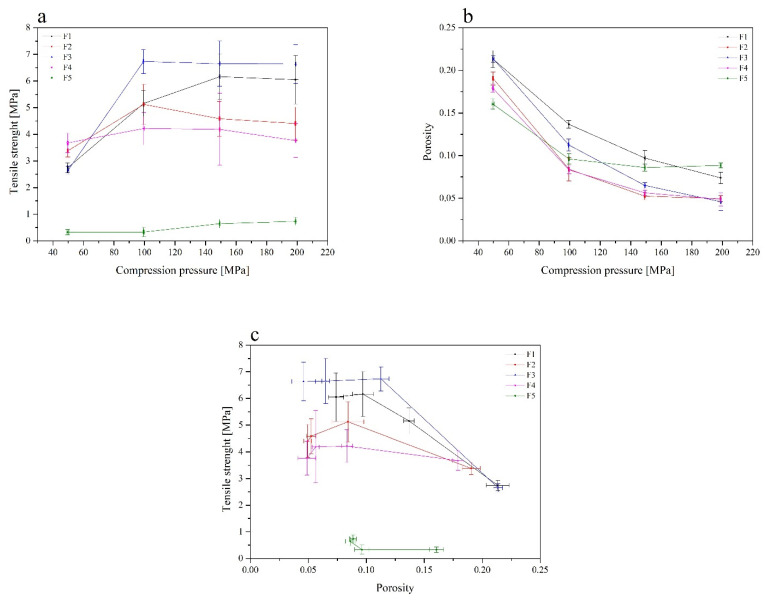
Tabletability (**a**), compressibility (**b**) and compactibility (**c**) profiles of the RSV cyclodextrin systems (F1-RSV/α-CD 1:1 DM, F2-RSV/β-CD 1:1 DM, F3-RSV/γ-CD 1:1 DM, F4-RSV/HP-β-CD 1:1 DM, F5-RSV/HP-β-CD 1:1 Evap).

**Figure 7 pharmaceutics-13-00417-f007:**
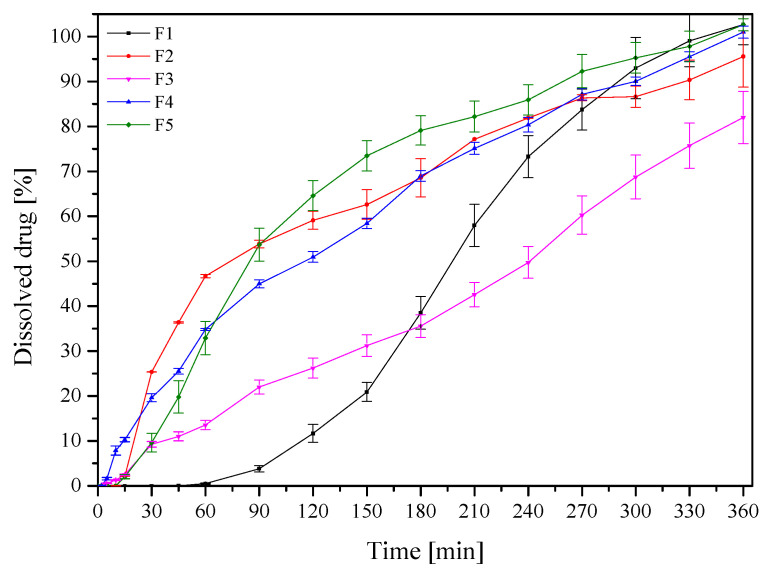
Dissolution profiles of the RSV/CD tablets (F1–F5) with the compression pressure 150 MPa.

**Figure 8 pharmaceutics-13-00417-f008:**
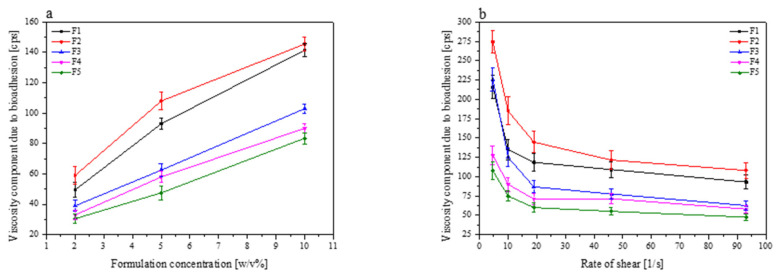
The effect of the polymer concentration (2–10%) on the viscosity component due to bioadhesion for formulations F1–F5 at a rate of shear 93 1/s (**a**), and the effect of the rate of shear on the viscosity component due to bioadhesion for 5% of formulations F1–F5 (**b**).

**Figure 9 pharmaceutics-13-00417-f009:**
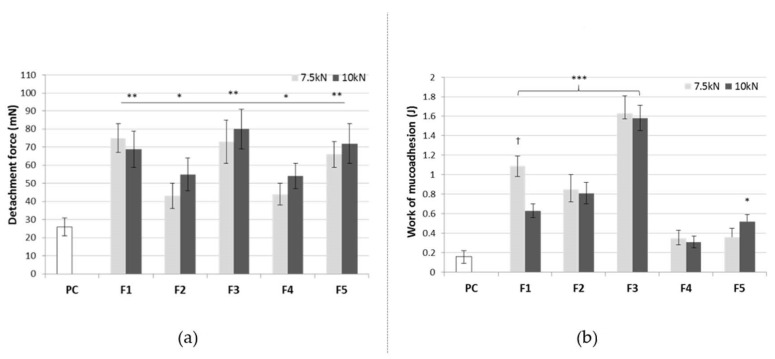
Maximum detachment force (**a**) and work of mucoadhesion (**b**) of tablets with resveratrol (F1–F5) compressed at different compaction forces (7.5 or 10 kN). Values are expressed as a mean ± SD; * represents significant differences with *p* ≤ 0.05, ** with *p* ≤ 0.01 while *** *p* ≤ 0.001 in comparison to control (cellulose paper, PC); † symbolizes significant differences with *p* ≤ 0.05 between the formulation compressed at 7.5 or 10 kN corresponding to the compression pressure 150 and 200 MPa, respectively.

**Table 1 pharmaceutics-13-00417-t001:** The composition of tablet formulations.

	F1	F2	F3	F4	F5
Complexation method	DM	DM	DM	DM	Evap
	Content (mg) of compounds in one tablet				
RSV	20.0	20.0	20.0	20.0	20.0
α-CD	85.0	-	-	-	-
β-CD	-	99.0	-	-	-
γ-CD	-	-	128.0	-	-
HP-β-CD	-	-	-	114.0	114.0
xanthan gum	10.5	11.9	14.8	13.4	13.4
magnesium stearate	1.2	1.3	1.6	1.5	1.5
sum	116.7	132.2	164.4	148.9	148.9

**Table 2 pharmaceutics-13-00417-t002:** Residence time (expressed in min) of tablets with resveratrol F1–F5 differed in compaction force to the porcine buccal mucosa (*n* = 3; median).

Compression Pressure	F1	F2	F3	F4	F5
150 MPa	>180	>180	120	10	>180
200 MPa	>180	>180	100	10	>180

## Data Availability

The data presented in this study are available through whole manuscript and Appendix A.

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
