# Peer review of "Buccal Resveratrol Delivery System as a Potential New Concept for the Periodontitis Treatment"

_pharmaceutics, 2021, doi:10.3390/pharmaceutics13030417_

Round 1

Reviewer 1 Report

The present manuscript deals with the formulation of a mucoadhesive tablet formulation for  buccal administration of the antioxidant compound resveratrol complexed with cyclodexstrins. The manuscript is readable and, the experimental studies are adequate, including both pre-formulative and formulative studies. However, some improvements are required for manuscript pubblication.

In the introduction, it is not clear the relationship between resveratrol and periodontitis. The authors reported in line 103-105 "blocks neutrophil.....reduce periodontitis", but there is no references about that and it is not clear if it is a common medical practice to use resveratrol. Moreover, it is not clear which are the advantages of formulating a buccal drug dosage form with the respect to an oral drug dosage form in relation to resveratrol and periodontitis. This aspects should be clarified since the use of the presented formulation for periodontitis is claimed in the title of the manuscript. 

The authors should explain better which is the rationale of including resveratrol inside cyclodextrins in relation to the presented mucoadhesive tablet formulation and its implications in the presented results. It would also better to formulate resveratrol/xanthan gum tablets without cyclodextrins for comparisons. 

English must be carefully revised since there are some mistakes throughout the manuscript.

DPPH and CUPRAC abbreviations should be defined

Line 284 "mucin-chitosan bioadhesive bond strenght" ????

Line 402 RES or RSV???

Line 504-514 The description of the release study must be improved.  Why xanthan gum was selected for the tablet formulation? In which way, the percentage of xantan gum and the presence of cyclodextrins influences the release? On which scientific evidence, the authors have affirmed that "xanthan gum did not influence the RSV dissolution rate" and "HP-beta-CD can act as a controlled-release excipient". Release profiles shown in Figure 7 differ from each other and they must be adequately discussed. F1 release profile is quite uncommon.

Caption of Figure 8 What is "component of bioadhesion"? There is no error bars in Figure 8.

Figure 9 There is no error bars. No statical comparision are reliable without errors and data cannot be presented only as median for statistical analysis. The statistical test performed is not reported.

Conclusions are very poor and general. They must be tailored to the findings presented in the manuscript.

Author Response

Manuscript number: pharmaceutics-1126238 

Manuscript title: A new approach of periodontitis treatment based on the buccal resveratrol delivery system

Author: Magdalena Paczkowska-Walendowska, Jakub DvoÅ™ák, Natalia Rosiak, Ewa Tykarska, Emilia SzymaÅ„ska, Katarzyna Winnicka, Marek A. RuchaÅ‚a, Judyta Cielecka-Piontek Dear Ms. Jelena Malešev,EditorPharmaceutics We would like to kindly thank You for your and Reviewer’s thorough review that helped us to improve our paper. We have taken into account all the suggestions and have made the necessary changes. Our responses are as follows:  

Comments from the Editors and Reviewer:
-Reviewer 1

The present manuscript deals with the formulation of a mucoadhesive tablet formulation for  buccal administration of the antioxidant compound resveratrol complexed with cyclodexstrins. The manuscript is readable and, the experimental studies are adequate, including both pre-formulative and formulative studies. However, some improvements are required for manuscript pubblication.

In the introduction, it is not clear the relationship between resveratrol and periodontitis. The authors reported in line 103-105 "blocks neutrophil.....reduce periodontitis", but there is no references about that and it is not clear if it is a common medical practice to use resveratrol. Moreover, it is not clear which are the advantages of formulating a buccal drug dosage form with the respect to an oral drug dosage form in relation to resveratrol and periodontitis. This aspects should be clarified since the use of the presented formulation for periodontitis is claimed in the title of the manuscript.

Response: Authors have rewritten the introduction section to clarified basics of periodontitis as well as hypothesize the use of resveratrol in its treatment [pages 1-3].        

The authors should explain better which is the rationale of including resveratrol inside cyclodextrins in relation to the presented mucoadhesive tablet formulation and its implications in the presented results. It would also better to formulate resveratrol/xanthan gum tablets without cyclodextrins for comparisons.

Response: The preparation of resveratrol systems with cyclodextrins at the preformulation stage was aimed at improving the physicochemical properties of the resveratrol. This was achieved in the context of increasing the dissolution rate of resveratrol from the cyclodextrin system. Only the solute is likely to work at the site of administration. Increasing the dissolution rate is extremely important, in particular for poorly soluble substances belonging to the 2nd BCS class, to which resveratrol belongs. There are few scientific reports on the mucoadhesive properties of cyclodextrins themselves. However, their use in the proposed system with resveratrol was not intended to increase the mucoadhesive properties, but to improve the physicochemical properties of the initial resveratrol. Xanthan gum was used to ensure mucoadhesive properties in the buccal tablets. More details were added throughout the revised version of manuscript [pages 8 and 12].

English must be carefully revised since there are some mistakes throughout the manuscript.

Response: We thank you for the note. All Authors carefully read all manuscript and corrected mistakes throughout the revised version of manuscript.

DPPH and CUPRAC abbreviations should be defined

Response: The DPPH and CUPRAC abbreviations were defined in Section 2.2.3.3 [page 6].

Line 284 "mucin-chitosan bioadhesive bond strenght" ????

Response: The method indicated in the section 2.3.4.1, described for the first time by Hassan and Gallo, is based on simple rheological measurements, i.e. the viscosity. During the test, 20% mucin solution is added to the polymer/test sample solution. The energy of the physical and chemical bonds of the mucin-polymer interaction can be transformed into mechanical energy or work. This work, which causes the rearrangements of the macromolecules, is the basis of the change in viscosity. The resulting interactions of the polymer with mucin will increase the viscosity of the final solution, i.e. the greater the interaction (potential adherence to the mucosa), the higher the viscosity value is observed. The results obtained for the powder systems using described rheological method have been confirmed and are consistent with those obtained for buccal tablets in the determination of the ex vivo mucoadhesive properties (maximum detachment force and work of mucoadhesion) as well as the residence time.

Line 402 RES or RSV???

Response: The spelling error has been corrected [page 12].

Line 504-514 The description of the release study must be improved.  Why xanthan gum was selected for the tablet formulation? In which way, the percentage of xantan gum and the presence of cyclodextrins influences the release? On which scientific evidence, the authors have affirmed that "xanthan gum did not influence the RSV dissolution rate" and "HP-beta-CD can act as a controlled-release excipient". Release profiles shown in Figure 7 differ from each other and they must be adequately discussed. F1 release profile is quite uncommon.

Response: Authors apologize very much for using some mental shortcuts that have proved to be misinterpreted. Of course xanthan gum influence the RSV dissolution rate, and play role as prolonged dissolution agent. It was confirmed by preparing tablets without cyclodextrins and shapes of obtained dissolution profiles were very similar to those presented in the manuscript in Figure 7. Moreover, all cyclodextrins can play as controlled-release excipient was is evident in Figure 4 where dissolution profiles for powder systems are presented. Therefore, the prolonged release of resveratrol from buccal tablets is the result of the use of cyclodextrin and xanthan gum in tablets composition. Additional comment has been placed in the revised version of manuscript [page 14].

Dissolution profiles of all formulation fitted to zero-order kinetic as well as Higuchi model. We could fully agree that dissolution profile shape of formulation 1 seems different than other but still fitted to the same kinetic models. All mathematical models results are presented in supplementary material.

Caption of Figure 8 What is "component of bioadhesion"? There is no error bars in Figure 8.

Response: Viscosity of a dispersion containing mucin and a bioadhesive system has to be considered as the results of the contribution of different components, such as the viscosities of bioadhesive polymer and mucin and the viscosity component due to bioadhesion (ηb). To be more precise, the statement "component of bioadhesion" in Figure 8 has been replaced with "viscosity component due to bioadhesion". Error bars in Figure 8 have been added [page 17].

Figure 9 There is no error bars. No statical comparision are reliable without errors and data cannot be presented only as median for statistical analysis. The statistical test performed is not reported.

Response: According to reviewer suggestion, the results form mucoadhesive studies were displayed as mean values with standard deviations of maximum detachment force and work of mucoadhesion [Figure 9, page 10]. Proper information regarding the statistical analysis was included in the section 2.3.4.2 [pages 8-9] as follow: “Quantitative variables were expressed as mean ± standard deviation (S.D.) by MS Excel software. Measurements were considered significant at p<0.05. Results from mucoadhesive studies were evaluated statistically by non-parametric Kruskal-Wallis test followed with Dunn-Bonferroni post hoc method with using Statistica 12.0 software. Shapiro-Wilk test was used to check the normality of data distribution.”

Conclusions are very poor and general. They must be tailored to the findings presented in the manuscript.

Response: Conclusions have been rewritten. We hope that the conclusions in their current form reflect the obtained research results more closely [page 19].

Reviewer 2 Report

Abstract:

The aim of the research was to investigate the possibility to formulate mucoadhesive cyclodextrin- and xanthan gum- based buccal tablets in order to increase  the solubility of resveratrol and to eliminate bypass enterohepatic metabolism. Systems of resveratrol with α-cyclodextrin (α-CD), β-cyclodextrin (β-CD), γ-cyclodextrin (γ-CD) and 2-hydroxypro- pyl-β-cyclodextrin (HP-β-CD) prepared by the dry mixing method (ratio 1:1) were selected for the  of tablets where xanthan gum was used as a mucoadhesive agent. They were identified on the basis  of PXRD, FT-IR analysis. Tablets F1 (with α-CD), F2 (with β-CD) and F3 (with γ-CD) were characterized by the highest compactibility as well as by favourable mucoadhesive properties. Resveratrol release from these tablets was delayed and controlled by diffusion. The tablets prepared in the course of this study appear to constitute promising resveratrol delivery systems and are recommended to increase the effectiveness of the treatment in many diseases, particularly periodontitis.

General comments

The paper deals with the development  of   resveratrol containing mucoadhesive tablets for the  treatment of periodontitis. The authors have performed intensive preformulation and formulation studies the majority of which are neither  justified nor direcly related to the intended goal.

The experimental design is too bulky and some tests are not relevant

The treatment of periodontitis ideally requires that the delivery system is inserted in the pocket and releases the active directly to the site of action.

The use of mucoadhesive system could help absorption of the active provided it is applied to the target tissue. Besided that, even though the increase concentration of resveratrol in saliva could lead to an increased active concentration at the site of action, no proof of evidence that this contributes to periodontitis  treatment is provided in this paper.

The authors should restrict the goal of the study. Also some tests performed  are not relevant to the scope. For instance the rhelogical test for mucoadhesion is not relevant for assesssing the mucoahesive properties of tablets. The dissolution method used is most relevant to simulate release in the gi tract rather than in the mouth. Volume and hydrodynamic conditions are not simulating the buccal environment. The fitting of dissolution data is also not relevant.  F1 and F2 parameters are generally used for regulatory purpose rather . See also specifc comments

Also the permability test is not relevant to transmucosal absorption and specialized ex-vivo models are available for that.

Specific comments

F1 and F2 paramters are generally used for regulatory purpose rather than for comparing dissolution profile.

The equation used for permeability should has a bibliographic reference.

The artificial membrane used for PAMPA is not suitable to assess the possible influence of cyclodextrin as absorption promoters.

Author Response

Manuscript number: pharmaceutics-1126238 

Manuscript title: A new approach of periodontitis treatment based on the buccal resveratrol delivery system

Author: Magdalena Paczkowska-Walendowska, Jakub DvoÅ™ák, Natalia Rosiak, Ewa Tykarska, Emilia SzymaÅ„ska, Katarzyna Winnicka, Marek A. RuchaÅ‚a, Judyta Cielecka-Piontek Dear Ms. Jelena Malešev,EditorPharmaceutics We would like to kindly thank You for your and Reviewer’s thorough review that helped us to improve our paper. We have taken into account all the suggestions and have made the necessary changes. Our responses are as follows:

Comments from the Editors and Reviewer:
-Reviewer 2

Abstract:

The aim of the research was to investigate the possibility to formulate mucoadhesive cyclodextrin- and xanthan gum- based buccal tablets in order to increase  the solubility of resveratrol and to eliminate bypass enterohepatic metabolism. Systems of resveratrol with α-cyclodextrin (α-CD), β-cyclodextrin (β-CD), γ-cyclodextrin (γ-CD) and 2-hydroxypro- pyl-β-cyclodextrin (HP-β-CD) prepared by the dry mixing method (ratio 1:1) were selected for the  of tablets where xanthan gum was used as a mucoadhesive agent. They were identified on the basis  of PXRD, FT-IR analysis. Tablets F1 (with α-CD), F2 (with β-CD) and F3 (with γ-CD) were characterized by the highest compactibility as well as by favourable mucoadhesive properties. Resveratrol release from these tablets was delayed and controlled by diffusion. The tablets prepared in the course of this study appear to constitute promising resveratrol delivery systems and are recommended to increase the effectiveness of the treatment in many diseases, particularly periodontitis.

General comments

The paper deals with the development of resveratrol containing mucoadhesive tablets for the treatment of periodontitis. The authors have performed intensive preformulation and formulation studies the majority of which are neither  justified nor direcly related to the intended goal.

Response: Authors extended the introduction indicating oxidative stress as one of the causes of periodontal diseases. Resveratrol is a well-known antioxidant that can function in this form. In addition, anti-inflammatory properties along with the mechanism of action have been well known and described in the literature. The aim of the research was not to prove the biological properties of resveratrol itself, but to use an appropriate pharmaceutical form that can be used in the treatment of periodontal diseases. The proposed studies of antioxidant activity were only aimed at confirming the activity of resveratrol in the presence of excipients such as cyclodextrins or xanthan gum as mucoadhesive agent. In addition, the use of a mucoadhesive formulation within the oral cavity is justified for the prolonged contact of the active substance in the affected area. Hence our research is directed to the journal based on pharmaceutical technology.

The experimental design is too bulky and some tests are not relevant

Response: The intention of the Authors, and at the same time the aim of the study, was to present both the part concerning preformulation research, including obtaining resveratrol systems with different cyclodextrins using different available methods, carrying out their identification and examining the physicochemical properties in order to select the systems with the best properties; but also carrying out formulation tests with the use of systems selected in the first part. In the opinion of the authors, separating both parts into two separate publications is not justified, as there are reports of the preparation of other resveratrol systems with cyclodextrins. The Authors tried to guide the reader step by step through the successive stages of both preformulation and formulation research. The authors hope that the changes made to the manuscript will prove helpful in reading the paper.

The treatment of periodontitis ideally requires that the delivery system is inserted in the pocket and releases the active directly to the site of action.

Response: The Authors agree with the reviewer's opinion. The proposed form of a mucoadhesive tablet aims to maintain the formulation (by creating bonds between the mucin and the polymer) at the application site, i.e. the affected area. The proposed qualitative and quantitative composition of excipients successfully ensures the maintenance of the mucosa on the surface, while maintaining the release level of active substances ensuring effectiveness. Increasing the amount of mucoadhesive polymer for prolonged contact with the mucosa could result in a decrease in the release rate of the active compounds; therefore, the composition of the proposed formulation is the result of many studies, both at the preformulation and formulation stage. Introducing the formulation into the teeth pocket and not on the surface of the mucosa requires the intervention of a dentist, and the idea of the proposed formulation is the ease of its application by the patient.

The use of mucoadhesive system could help absorption of the active provided it is applied to the target tissue. Besided that, even though the increase concentration of resveratrol in saliva could lead to an increased active concentration at the site of action, no proof of evidence that this contributes to periodontitis  treatment is provided in this paper.

Response: The use of resveratrol in the treatment of periodontitis the Authors based on the documented antioxidant, anti-inflammatory and antibacterial properties of the molecule. The Authors presented studies confirming the antioxidant activity of resveratrol in the form of a powder, as well as in the final composition of the formulation. Therefore, the main goal of the research was to present the possibility of using resveratrol in the mucoadhesive form, while the use in periodontitis is an indication resulting from the activity of resveratrol as well as the selected pharmaceutical form. The authors are aware of the need to conduct further clinical trials to confirm the effectiveness of the product.

The authors should restrict the goal of the study. Also some tests performed  are not relevant to the scope. For instance the rhelogical test for mucoadhesion is not relevant for assesssing the mucoahesive properties of tablets. The dissolution method used is most relevant to simulate release in the gi tract rather than in the mouth. Volume and hydrodynamic conditions are not simulating the buccal environment. The fitting of dissolution data is also not relevant.  F1 and F2 parameters are generally used for regulatory purpose rather . See also specifc comments

Response: The use of rheological tests was to confirm the correctness of the use of excipients in the formulation. The Authors fully agree with the reviewer's opinion that rheological tests are not dedicated to the study of tablets, but in the opinion of the Authors, they are equally valuable information about the mucoadhesive properties of the proposed combination. The use of rheological research is widely described in the literature, please see J Control Release 1998, 50(1-3), 167-178, Eur J Pharm Biopharm 2013, 85(3),  843-853 as well as J Pharm Biomed 2018, 156, 232-238.

Authors performed a dissolution rate test for both powder systems (Section 2.2.3.1) and buccal tablets (Section 2.3.3). Probably the description of the methodology for buccal tablets was too poor and the parameters used could be misread. Tablets were tested at an orally appropriate pH with a small volume of medium using generally recognized methodology due to the lack of pharmacopoeial guidelines for this pharmaceutical form. Authors supplemented the revised version of manuscript with a detailed description of this dissolution study of tablets.

F1 and F2 parameters are used to compare the release profile of the active substance from the reference and generic drugs, Authors fully agree with Reviewer’s comment. But, many years ago, when we started our dissolution research in our team, we started to consider comparing the dissolution profiles with powder systems. There is no dedicated method for such a statistical survey; hence the borrowing of the method from the study of pharmaceutical forms. We are aware that this method has some limitations, but we successfully use it as a statistical tool to compare release profiles from powder systems (please see our previous papers: J Incl Phenom Macrocycl Chem 2018, 91, 149-159; Scientific Reports 2018, 8, 16184; Plos One 2019, https://doi.org/10.1371/journal.pone.0210694; Int J Pharm 2020, 581, 119294; J Clin Med 2020, 9, 1208; Pharmaceutics 2020, 12, 634). The use of a statistical method at the preformulation stage is an important indication of whether and how the addition of excipients and/or sample preparation methods affect the dissolution rate of active substances.

Also the permability test is not relevant to transmucosal absorption and specialized ex-vivo models are available for that.

Response: Authors fully agree with Reviewer’s comment that on the market there are some ex vivo transmucosal permeation tests. Unfortunately, the Authors do not have access to the said test, and the aim of the research was not to accurately assess the penetration of resveratrol through the oral mucosa, but to maintain its high concentration on the mucosa surface, the use of the PAMPA model at the preformulation stage is also advisable. Based on literature data, Khdair and co-workers linear relationships between the permeation of carvedilol through PAMPA model and rabbit and porcine mucosa (J Drug Delivery Sci Technol 2013, 23 603-609; Eur J Pharm Sci 2018, 119, 2018, 219-233). Moreover, the use of the PAMPA model was not intended to assess cyclodextrins as absorption promoters, since the Authors did not care about the systemic effect of the compound. The purpose of using the pharmaceutical formulation was to maintain resveratrol on the surface of the oral mucosa, which was proved by using the PAMPA model. In fact, resveratrol belonging to 2nd class of BCS is characterized by high permeability, while the use of cyclodextrins at the preformulation stage did not result in a statistically significant increase in resveratrol permeability. In addition, the PAMPA model is recommended for screening tests on a large number of trials in order to save time and money. Certainly, ex vivo tests are much more expensive than a simple in-vitro test. Based on that, the Authors assume that the use of the PAMPA model at an early stage of preformulation studies is advisable.

Specific comments

F1 and F2 paramters are generally used for regulatory purpose rather than for comparing dissolution profile.

Response: Please look into our response above.

The equation used for permeability should has a bibliographic reference.

Response: A permeability equation reference was added as number 35 in the revised version of manuscript.

The artificial membrane used for PAMPA is not suitable to assess the possible influence of cyclodextrin as absorption promoters.

Response: The use of the PAMPA model was not intended to assess cyclodextrins as absorption promoters, since the Authors did not care about the systemic effect of the compound. The purpose of using the pharmaceutical formulation was to maintain resveratrol on the surface of the oral mucosa, which was proved by using the PAMPA model. In fact, resveratrol belonging to 2nd class of BCS is characterized by high permeability, while the use of cyclodextrins at the preformulation stage did not result in a statistically significant increase in resveratrol permeability.

Reviewer 3 Report

Overall, the manuscript is well-written and of interest to the readers of Pharmaceutics. However, there are few comments that need revision before publication: 

  1. You have stated in the introduction that resveratrol has a poor water solubility. Up to what extent, the different cyclodextrin enhances the solubility of resveratrol in aqueous media? I think this work is missing solubility studies. Please check this paper, New amphotericin B-gamma cyclodextrin formulation for topical use with synergistic activity against diverse fungal species and Leishmania spp. The stability constant and complexation efficiency should be calculated at least with the combination that works the best. 
  2. Are you interested in obtaining an amorphous solid dispersion? 
  3. Dissolution studies have been performed with powder inside gelatin capsules, don't you think to test the buccal tablets would be more representative?
  4. Statistical analysis is missing. Please, check it in all your results needed.
  5. Which is the formulation with the better performance?
  6.  discussion is missing, please compare your results with those obtained by other authors. 

Author Response

Manuscript number: pharmaceutics-1126238 

Manuscript title: A new approach of periodontitis treatment based on the buccal resveratrol delivery system

Author: Magdalena Paczkowska-Walendowska, Jakub DvoÅ™ák, Natalia Rosiak, Ewa Tykarska, Emilia SzymaÅ„ska, Katarzyna Winnicka, Marek A. RuchaÅ‚a, Judyta Cielecka-Piontek 

Dear Ms. Jelena Malešev,

Editor

Pharmaceutics 

We would like to kindly thank You for your and Reviewer’s thorough review that helped us to improve our paper. We have taken into account all the suggestions and have made the necessary changes. Our responses are as follows:

Comments from the Editors and Reviewer:
-Reviewer 3

Overall, the manuscript is well-written and of interest to the readers of Pharmaceutics. However, there are few comments that need revision before publication:

  1. You have stated in the introduction that resveratrol has a poor water solubility. Up to what extent, the different cyclodextrin enhances the solubility of resveratrol in aqueous media? I think this work is missing solubility studies. Please check this paper, New amphotericin B-gamma cyclodextrin formulation for topical use with synergistic activity against diverse fungal species and Leishmania spp. The stability constant and complexation efficiency should be calculated at least with the combination that works the best.

Response: Based on literature data we know that all used cyclodextrin (α-CD, β-CD, γ-CD, HP-β-CD)  can increased the resveratrol water solubility. It was measured by phase-solubility studies described in J Incl Phenom Macrocycl Chem 2006, 55, 279-287 and Food Chemistry 2009, 113, 17-20. Our goal was to confirm that by using our preparation method we would obtained such increase in solubility. We can confirm that by using powder dissolution studies, where 3-fold increasing in resveratrol solubility was observed.

  1. Are you interested in obtaining an amorphous solid dispersion?

Response: Obtaining an amorphous solid dispersion wasn’t our primary goal. From our previous experience, we know that the methodology used to prepare cyclodextrin powder systems, including the use of amorphous HPβCD, may lead to the formation of amorphous systems. The aim of our research was to check how the applied methodology and the type of cyclodextin would affect the physicochemical properties of the initial resveratrol. While expecting an interaction between the resveratrol and cyclodextrin molecule, we used both PXRD to check for amorphous scattering and FT-IR analysis to determine the types of interactions. Moreover, our main goal was to assess the changes in the physicochemical properties of resveratrol as a result of interaction with the cyclodextrin molecule.

  1. Dissolution studies have been performed with powder inside gelatin capsules, don't you think to test the buccal tablets would be more representative?

Response: Authors performed a dissolution rate test for both powder systems (Section 2.2.3.1) and buccal tablets (Section 2.3.3). Probably the description of the methodology for buccal tablets was too poor and the parameters used could be misread. Authors supplemented the revised version of manuscript with a detailed description of this dissolution study of tablets. Dissolution of resveratrol from powder systems is shown in Figure 4, while tablets dissolution in Figure 7.

  1. Statistical analysis is missing. Please, check it in all your results needed.

Response: The dissolution study from powder systems was supplemented with a statistical evaluation with the use of difference (F1) and similarity (F2) factors [Section 2.2.3.1]. The same method was used to assess the similarity of the solubility profiles of resveratrol with buccal tablets – the missing information was added into revised version of manuscript [Section 2.3.3, page 7].

Permeability studies using PAMPA model was supplemented with a statistical evaluation with the use of ANOVA test [Section 2.2.3.2]. ANOVA test was also used in antioxidant activity assays - the missing information was added into revised version of manuscript [Section 2.2.3.3, pages 5-6].

In case of mucoadhesive properties quantitative variables were expressed as mean ± standard deviation (S.D.) by MS Excel software. Measurements were considered significant at p<0.05. Results from mucoadhesive studies were evaluated statistically by non-parametric Kruskal-Wallis test followed with Dunn-Bonferroni post hoc method with using Statistica 12.0 software. Shapiro-Wilk test was used to check the normality of data distribution [Section 2.3.4.2 and 2.3.4.4, pages 8-9].

  1. Which is the formulation with the better performance?

Response: Based on all the obtained results, formulation 2 containing resveratrol with β-cyclodextrin turned out to be the most valuable. This formulation exhibited good tabletability, compressibility, as well as good compactibility, not the highest values. As a result, a quick release of resveratrol from buccal tablets was obtained. In addition, this formulation showed good mucoadhesive properties, allowing the tablet to remain at the site of administration for 3 hours. This time, in turn, ensures that the resveratrol is released from the tablet. In fact, there was no fragment in the paper discussing the choice of the best formulations. the missing discussion was added to the revised version of manuscript [page 19].

  1. discussion is missing, please compare your results with those obtained by other authors.

Response: In case of formulation studies in the literature data there is no papers about resveratrol-cyclodextrin tableting process so it was difficult to discuss directly previous results with those obtained in our work, but discussion about the use of excipients in an analogous formulation is included. In other aspects, an additional discussion of the results was included in the work test. Please take a look at the revised version of manuscript throughout the section 3.

Round 2

Reviewer 1 Report

The comments have been addressed by the authors and the manuscript is suitable for pubblication

Author Response

Comments from the Editors and Reviewer:
-Reviewer 1

The comments have been addressed by the authors and the manuscript is suitable for pubblication

Response: The Authors are grateful to the Reviewer for his/her appreciation of the contributions and responses to improving the manuscript.               

Reviewer 2 Report

The authors have not accepted the referee's suggestions to simplify the experimental design and have not made any significant changes to the previous script.

They have however amended the dissolution section and explained their goal. It appears clearly that reference to periodontitis is vague.

In order to publish the paper as such, it is suggested to change the title eliminating reference to periodontisis

Author Response

Comments from the Editors and Reviewer:
-Reviewer 2

Abstract:

The authors have not accepted the referee's suggestions to simplify the experimental design and have not made any significant changes to the previous script.

They have however amended the dissolution section and explained their goal. It appears clearly that reference to periodontitis is vague.

In order to publish the paper as such, it is suggested to change the title eliminating reference to periodontisis

Response: The Authors took into account the Reviewer's opinion and proposed a new title for the revised version of the manuscript..